# Marine Alkaloids with Anti-Inflammatory Activity: Current Knowledge and Future Perspectives

**DOI:** 10.3390/md18030147

**Published:** 2020-03-02

**Authors:** Cássio R. M. Souza, Wallace P. Bezerra, Janeusa T. Souto

**Affiliations:** Department of Microbiology and Parasitology, Federal University of Rio Grande do Norte, Avenida Senador Salgado Filho, BR 101, Campus Universitário, Lagoa Nova, Natal RN 59078-900, Brazil

**Keywords:** marine alkaloids, anti-inflammatory, marine algae, bioprospecting of marine natural compounds

## Abstract

Alkaloids are nitrogenous compounds with various biological activities. Alkaloids with anti-inflammatory activity are commonly found in terrestrial plants, but there are few records of the identification and characterization of the activity of these compounds in marine organisms such as fungi, bacteria, sponges, ascidians, and cnidarians. Seaweed are a source of several already elucidated bioactive compounds, but few studies have described and characterized the activity of seaweed alkaloids with anti-inflammatory properties. In this review, we have gathered the current knowledge about marine alkaloids with anti-inflammatory activity and suggest future perspectives for the study and bioprospecting of these compounds.

## 1. Introduction

Since the early days of medicine, civilizations have used plants, herbs, and their derivatives for the treatment of numerous diseases. According to the World Health Organization, the use of traditional and complementary medicines, among which herbal medicines, is extensive and involves about 80% of member states’ population [1]. The scientific community has turned to medicinal plants as one of the largest sources of biologically active substances [2] in order to identify, characterize, and understand the mechanisms of action of numerous compounds found in these species. Many studies have shown the anti-inflammatory potential of substances found in plants and seaweeds that act on major inflammation control pathways [3,4]; however, it appears that pharmacological activities have been elucidated for only a few of these species [5].

The ocean, which hosts the largest concentration of species on the planet, became the target of scientific interest around the middle of the 20th century. Since then, tens of thousands of new substances with pharmacological potential have been described [6]. Among all species living in the oceans, seaweed is a promising source of new bioactive and pharmacological compounds [7]. Seaweeds are photosynthetic autotrophic organisms classified into different taxonomic groups, which produce a range of chemically distinct compounds [8]. Some of these compounds are not present in plants and have bioactive potential, of interest for pharmaceutics, cosmetics, and nutrition [9,10,11,12]. Seaweeds are sources of many biochemical compounds such as lipids, vitamins, proteins, polysaccharides, fibers, minerals, and others [11,13,14,15,16]. In addition to these compounds, seaweed produce a number of secondary metabolites with remarkable biological activity, such as phenolic compounds, polysaccharides, carotenoids, lectins, steroids, polyketides, and many others [17]. Several studies have shown that seaweed contain a range of bioactive substances with diverse pharmacological potential, such as antiviral [18], antibiotic and antiendotoxic [19], antifungal [20], antiparasitic [21], antioxidant [22], anti-ageing [23], antinociceptive [24], anti-tumor [25], anti-diabetic [26], anti-inflammatory, and immunomodulatory [27,28] effects. Among all, the anti-inflammatory activity of compounds derived from seaweed is one of the largest bioprospecting areas in marine natural products.

Several studies have already demonstrated the activity of compounds of different chemical nature from those of seaweed in different experimental models, with a precise characterization of their structures, activities, and mechanisms of action. Compounds such as phenols and polyphenols, carotenoids, proteins and peptides, and polysaccharides have well-characterized anti-inflammatory activity [29]. In addition to these compounds, other classes of molecules have raised interest in this area of bioprospecting. Alkaloids, for example, are nitrogenous compounds derived from amino acids that have a wide range of biological activities [29,30]. Some chemical characteristics of these molecules, such as ionization profile and capacity to form stable salts, may indicate their higher potential as drug candidates, when compared with other compound classes. Around 95% of drugs are estimated to have an ionizable group (75% weak bases, 20% weak acids, and 5% non-ionic molecules, ampholytes, and alcohols) [31]. For a given compound to be considered drug-like (or even lead-like), its ability to ionize is pivotal, as ionization is important in various phases of the drug optimization process [32]. Ionizable groups are key to fine-tuning the aqueous solubility, or hydrophilicity [33], and CNS permeability [34] of a molecule and to enhancing its activity [35]. Because of their ionization profile, alkaloids (which are largely weak bases) are good drug candidates better than substances with a non-ionization profile. In addition, basic groups, such as amines, amides, amidine, and guanidine, can form salts in biological media. The incorporation of these groups yields molecules with low hydrophobicity. Many drugs containing basic groups perform their activities by ionic hydrogen bonds (when salts) or induced dipoles. In this case, alkaloids can form stable salts that have better pharmacokinetic properties than non-basic drugs [36].

Alkaloids from plants are an important class of molecules with anti-inflammatory activity [37,38], demonstrating inhibition of expression of several pro-inflammatory factors, such as cytokines, lipid mediators, histamine, and enzymes involved in the inflammatory response [39]. Although most commonly found in plants, alkaloids can also be obtained from marine organisms. Trabectedin, for example, is an alkaloid obtained from a marine ascidian with a well-documented antitumoral activity and actually used as a treatment against many types of cancer [40,41]. Another example is pyridoacridines, a class of marine alkaloids obtained from different organisms, for which many biological activities have already been identified and characterized [42]. Several studies have also described the ability of marine alkaloids to inhibit proinflammatory factors in in vitro and in vivo experimental models [43]. However, the characterization of the anti-inflammatory activity of seaweed alkaloids is still understudied and underdeveloped and thus requires great efforts in the field of bioprospecting marine natural products.

In this review, we have gathered the current knowledge about the anti-inflammatory activity of alkaloids from marine organisms. More specifically, we have focused on seaweed, reviewing their mechanisms of action and discussing the state of the art and future perspectives for identification and characterization studies involving these compounds.

## 2. Anti-Inflammatory Alkaloids of Marine Organisms: Sources and Bioactivity

Marine alkaloids originating from marine organisms are described in the literature, presenting different biological and pharmacological activities. Several studies describe the presence of alkaloids in *Alcyonacea* coral species, with antiviral, antitumor, antibiotic, and immunomodulatory activities [44]. Similarly, many sponge alkaloids have been identified, and their potential antibiotic, antifungal, antitumor, antileukemic, and antidepressant activities have been described [45,46,47,48,49]. Similar effects were observed for alkaloids obtained from genera of marine fungi [50]. Other biological activities described for marine alkaloids involve anti-parasitic, enzymatic, anti-serotonergic, and antiretroviral effects [51,52]. Despite all these potential therapeutic activities, bioprospecting of marine alkaloids with anti-inflammatory activity is still an open study field.

### 2.1. Marine Fungi- and Bacteria-Derived Alkaloids

Marine microorganisms such as fungi and bacteria are also sources of primary and secondary metabolites with anti-inflammatory activity. The major classes of molecules found in these organisms are peptides and proteins, lipids, polyketides, organic acids, and terpenoids [53,54,55]. Fungi, for example, are the source of a large number of marine alkaloids with known biological activity [50,56]. In addition, a large fraction of the nitrogenous compounds found in ascidians are alkaloids [57], and some of them exhibit anti-inflammatory activity.

Asperversiamides B (**1**), C (**2**), F (**3**), and G (**4**), indole alkaloids derived from the marine fungus *Aspergillus versicolor*, reduced NO levels by inhibiting iNOS activity in lipopolysaccharide (LPS)-stimulated RAW 264.7 macrophages [58]. Two *N,N’*-ketal quinazoline alkaloids, the enantiomers (±)-penicamide A [**5**: (+)-penicamide A; **6**: (-)-penicamide A], extracted from the fungus *Penicillium canescens* isolated from the ascidian *Styela plicata*, also reduced NO levels produced by LPS-stimulated macrophages [59]. The two diketopiperazine alkaloids 5-prenyl-dihydrovariecolorin F (**7**) and 5-prenyl-dihydrorubrumazine A (**8**), extracted from *Aspergillus* spp. found in seaweed, showed an inhibitory effect on iNOS and COX-2 activity, reducing NO and PGE_2_ levels produced by LPS-stimulated RAW 264.7 and BV2 cells [60]. Neoechinulin A (**9**), an indolic alkaloid extracted from marine fungi *Eurotium* spp., was able to reduce NO and PGE_2_ production by inhibiting iNOS and COX-2 expression and reduced the production of IL-1β and TNF-α in LPS-stimulated RAW 264.7 cells [61]. These anti-inflammatory effects were associated with the inhibition of IκB-α phosphorylation and degradation and the inhibition of NF-κB p65 subunit binding to nuclear DNA, which blocked the NF-κB pathway-mediated pro-inflammatory response. Neoechinulin A was also able to inhibit MAPK p38 phosphorylation, also involved in inducing a pro-inflammatory response.

Chaetoglobosin Fex (**10**), the chytocalasan-based alkaloid extracted from the fungus *Chaetomium globosum*, was able to reduce the production of TNF-α, IL-6, and MCP-1 and also inhibited IκB-α degradation, the translocation of the NF-κB p65 subunit to the cell nucleus, and the phosphorylation of p38, ERK1/2, and JNK MAPKs by LPS-stimulated macrophages [62]. This study also showed that Chaetoglobosin Fex was also able to reduce CD14 expression in LPS-stimulated cells. CD14 plays an important role in LPS recognition and in the initiation of a proinflammatory response by activation of the TRL4/MD-2 recognition complex [63]. Finally, the quinoline alkaloids actinoquinolines A (**11**) and B (**12**) extracted from bacteria *Streptomyces* spp. found in marine sediment were able to inhibit COX-1 and COX-2 activity in vitro [64]. Chemical structures of the alkaloids described in this section are shown in Figure 1.

### 2.2. Sponge-Derived Alkaloids

Most of the biologically active marine compounds already identified between 2001 and 2010 came from sponges [65] and, in the past decade, more than 1900 new bioactive compounds were obtained from these organisms, which thus appear as a major source of marine natural products [66,67,68,69,70,71,72]. Anti-inflammatory sponge compounds have an inhibitory effect on inflammatory mediators, such as cytokines and chemokines, and are able to modulate several enzymatic pathways involved in the synthesis of pro-inflammatory factors, such as COX-2, and cellular signaling pathways, such as the MAPK and NF-κB pathways [73,74,75,76]. Some studies also describe the anti-inflammatory activity of alkaloids from sponges and their derivatives, as discussed below.

Barettin (**13**), the brominated alkaloid extracted from the sponge *Geodia barretti*, showed in vitro anti-inflammatory and antioxidant activity, reducing both TNF-α and IL-1β levels in LPS-stimulated THP-1 cells [77]. In this study, barettin also showed a potent antioxidant effect, reducing lipid peroxidation (risk factor associated with chronic inflammation) in HepG2 cells. Halichlorine (**14**), an alkaloid extracted from the sponge *Halichondria okadai*, inhibited the expression of VCAM-1, ICAM-1, and E-selectin in LPS-stimulated aortic endothelial cells, inhibiting macrophage adhesion to cultured cell monolayers and exerting an anti-inflammatory effect associated with NF-κB pathway inhibition [78]. The pyrrole alkaloid (10*Z*)-debromohymenialdisine (**15**), extracted from a sponge of the genus *Stylissa*, was able to reduce the expression of IL-1β, IL-6, TNF-α, iNOS, and COX-2, reducing also the levels of NO and PGE_2_ in co-cultures of LPS-stimulated Caco-2 and THP-1 cells [79]. In this study, it was also observed that treatment of the cells with the alkaloid reduced the phosphorylation of p38, ERK1/2, JNK, and NF-κB p65 subunit in the cytoplasm, also reducing the translocation of phosphorylated p65 to the cell nucleus. Interestingly, in addition to the inhibitory effect on enzymes and proinflammatory factors, the alkaloid (10*Z*)-debromohymenialdisine induced an increase in HO-1 expression and the translocation of the transcription factor Nrf-2 to the cell nucleus; these are protein factors associated with both suppression of oxidative stress and inflammation in LPS-activated macrophages.

Stylissadines A (**16**) and B (**17**), two alkaloids obtained from the sponge *Stylissa flabellata*, showed a specific antagonistic effect on P2X7 receptors in THP-1 cells, with no signs of cytotoxicity to cells [80]. P2X7 receptors are ATP receptors involved in the activation of inflammatory response mechanisms, such as the activation of NLRP3 inflammasomes, the synthesis of proinflammatory cytokines and chemokines, the activation of MAPK and phospholipase C, and the activation of transcription factors involved in the expression of inflammatory genes, like NF-κB and NFATc1 [81]. The activity of these alkaloids as P2X7 receptor antagonists may therefore have broad therapeutic possibilities for inflammatory and immune-mediated diseases. The 6-bromoindole derivatives geobarettin B (**18**) and C (**19**), the 6-bromoindole alkaloids 6-bromoconicamin (**20**), and barettin (**13**), obtained from the sponge *G. barretti*, showed an anti-inflammatory effect on LPS-, TNF-α-, and IL-1β-stimulated dendritic cells, reducing IL-12p40 secretion [82]. Geobarettin C was also able to increase the levels of IL-10 secreted by these cells. In this study, dendritic cells (DCs) matured by treatment with geobarettin B and C were subsequently co-cultured with CD4^+^ T cells, and a reduction in IFN-γ levels produced by these T cells was observed. The data suggest an extended anti-inflammatory effect not only on innate immunity mechanisms but also on adaptive immune responses. A synthetic tricyclic guanidine (**21**), similar to guanidine alkaloids from marine sponges, also showed an inhibitory effect on IFN-γ and MCP-1/CCL2 levels in an in vitro model of macrophage infection by *Leishmania infantum* [83]. Chemical structures of the alkaloids described in this section are shown in Figure 2.

### 2.3. Other Invertebrate Animals as Sources of Marine Alkaloids

Alkaloids with anti-inflammatory activity have also been found in several other invertebrate marine organisms. The alkaloids tubastrine (**22**) and orthidines A (**23**), B (**24**), C (**25**), E (**26**), and F (**27**), isolated from the ascidian *Aplidium orthium*, were able to reduce superoxide synthesis in phorbol-12-myristate 13-acetate (PMA)-stimulated neutrophils in vitro and, in an in vivo study, reduced superoxide levels in a gouty arthritis model [84]. Also, tubastrine and orthidine F showed an inhibitory effect on neutrophil infiltration in this in vivo model. Ascidiathiazones A (**28**) and B (**29**), two thiazone-containing quinolinequinone alkaloids obtained from *Aplidium* spp., had a similar effect on superoxide production by PMA-stimulated neutrophils in vitro and in an in vivo murine gout model [85]. Kottamide D (**30**), the imidaloze-containing alkaloid obtained from the ascidian *Pycnoclavella kottae*, was also able to reduce superoxide synthesis by PMA and N-formylmethionyl-leucyl-phenylalanine (fMLP)-activated neutrophils in vitro [86]. Aqueous extracts from the anemones *Anemonia sulcata* and *Actinia equina*, whose major detected constituent was the methylpiridinium alkaloid homarine (**31**), were able to reduce NO and reactive oxygen species (ROS) production in LPS-stimulated RAW 264.7 cells and had an inhibitory effect on the activity of PLA_2_, one of the enzymes involved in the activation of the arachidonic acid pathway during the inflammatory response [87]. The compound 5α-iodozoanthenamine (**32**), a zoanthamine alkaloid from the cnidarian *Zoanthus kuroshio*, showed an anti-inflammatory effect on fMLP-stimulated neutrophils, reducing superoxide anion generation and elastase release by these cells [88]. The molecules 3-hydroxinorzoanthamine (**33**), norzoanthamine (**34**), and zoanthamine (**35**), additional zoanthamine alkaloids extracted from *Zoanthus* cf. *pulchellus*, promoted the reduction of ROS and NO produced by LPS-stimulated BV-2 cells [89]. Convolutamydine A (**36**), an oxindole alkaloid commonly found in marine bryozoans, and two synthetic analogous molecules, ISA147 and ISA003, were able to reduce leukocyte migration to the lesion site and the levels of IL-6, TNF-α, NO, and PGE_2_ in a model of carrageenan-induced inflammation in a subcutaneous air pouch [90]. Chemical structures of the alkaloids described in this section are shown in Figure 3.

## 3. Anti-Inflammatory Alkaloids in Marine Algae

As previously shown, seaweed are a source of numerous anti-inflammatory compounds already described. However, there are few studies on alkaloids. Most biologically active alkaloids described so far are from plants, while few studies describe the isolation and biological activity of seaweed alkaloids [29].

Among the algae with identified anti-inflammatory compounds, those of the genus *Caulerpa* are well studied and described in the literature [91,92,93]. Studies from our group showed anti-inflammatory activity of compounds extracted from green algae of the genus *Caulerpa*. *Caulerpa mexicana* aqueous and methanolic extracts were able to reduce IL-6, IL-12, and TNF-α production by LPS-stimulated macrophages and leukocyte migration in murine zimosan-induced peritonitis and air pouch inflammation models and decreased xylene-induced ear edema [27]. Subsequently, we observed the anti-inflammatory activity of a *C. mexicana* methanolic extract in a murine model of dextran sulfate sodium (DSS)-induced ulcerative colitis, with the attenuation of the clinical signs of the disease and a significant reduction of IFN-γ, IL-6, IL-12, IL-17A, and TNF-α levels, together with the preservation of the morphological structure of the colon and a reduction of inflammatory tissue infiltrates [28]. In fact, in another study, different extracts of *C. mexicana* and *Caulerpa sertularioides* showed anti-inflammatory activity in a murine model of carrageenan-induced peritonitis, reducing leukocyte migration to the lesion site [94].

Algae extracts of the genus *Caulerpa* are rich in caulerpin (**37**), an indolic alkaloid with proven anti-inflammatory activity. Caulerpin has been described in different species of the genus *Caulerpa*, such as *Caulerpa peltata*, *Caulerpa racemosa*, *Caulerpa cupressoides*, *Caulerpa paspaloides*, *Caulerpa prolifera*, *C. sertularioides*, *C. mexicana*, and *Caulerpa lentillifera*, besides being found in the red algae *Chondria armata* [95]. The analysis of *C. peltata* and *C. racemosa* ethanolic extracts showed caulerpin as one of the main products [96,97]. Other indolic alkaloids of the genus *Caulerpa* found in algae and already identified are racemosin A (**38**) [98], B (**39**) [99], and C (**40**), and caulersin (**41**) [100]. The first alkaloid showed protective activity on SH-SY5Y cell viability, the second one can give rise to alkylamide derivatives capable of inducing cell death in the breast cancer cell line MDA-MB-231, and the last two showed inhibitory activity on PTP1B; however, anti-inflammatory activity of these compounds has not been registered to date. On the other hand, the anti-inflammatory activity of caulerpin has been described by some studies. Caulerpin treatment was able to decrease the plasma extravasation in a murine model of capsaicin-induced ear edema and reduce total leukocyte migration and neutrophil migration in a murine model of carrageenan-induced peritonitis [101]. We also demonstrated the anti-inflammatory effect of caulerpin in a murine model of DSS-induced ulcerative colitis. Treatment of the animals with caulerpin attenuated the clinical signs of the disease, reduced inflammatory infiltrates and the levels of the proinflammatory cytokines IL-6, IL-12, TNF-α, and IFN-γ, and increased the levels of the anti-inflammatory cytokine IL-10 in the colon of the affected animals. Caulerpin treatment also reduced NF-κB p65 expression in the affected tissue, suggesting a central modulating effect on NF-κB activation [102].

Red algae of the genus *Gracilaria* have also been described as important sources of anti-inflammatory compounds and alkaloids with elucidated biological activity [103]. The aqueous extract of *Gracilaria tenuistipitata* showed anti-inflammatory activity in an in vitro Hepatitis C Virus (HCV)-induced inflammation model [104]. Treatment of HCV-infected cells with this extract was able to inhibit COX-2 activity and PGE_2_ synthesis, as well as NF-κB p65 translocation to the cell nucleus and TNF-α, IL-1β, and iNOS gene expression. On the other hand, a *Gracilaria changii* methanolic extract reduced TNF-α levels and TNF-α and IL-6 gene expression in PMA-stimulated U937 cells [105]. An azocinyl morpholinone alkaloid (**42**) extracted from *Gracilaria opuntia* showed anti-inflammatory activity in a murine model of carrageenan-induced paw edema, reducing edema formation by 6 h and exhibiting a selective inhibitory effect on COX-2 and 5-LOX activity [106]. Other red algae of the genus *Laurencia* are also sources of alkaloids with known biological activities and already elucidated antimicrobial potential [107,108,109]. On the other hand, a methanol/dichloromethane extract of *Laurencia obtusa* showed an anti-inflammatory effect on LPS-stimulated THP-1 cells, reducing TNF-α production by these cells, and decreased in vivo the inflammatory exudate in a murine model of carrageenan-induced paw edema [110]. The anti-inflammatory effect of the extract was attributed to the presence of secondary metabolites such as alkaloids and terpenoids. Chemical structures of the alkaloids described in this section are shown in Figure 4.

## 4. Bioprospecting of Marine Anti-Inflammatory Alkaloids and Future Perspectives

The marine biome is currently one of the largest sources of biologically active compounds, offering a great possibility of bioprospecting for new pharmacological treatments [6]. Micro- and macroalgae are organisms with great potential for bioprospecting marine natural products and are rich sources of compounds with already characterized antimicrobial, antitumor, anticoagulant, and anti-inflammatory activity [111,112,113,114]. Among these compounds, alkaloid have great biological potential. Despite this potential, until the last decade, no alkaloid identified from seaweed was used as a therapeutic resource in modern medicine [115].

Many alkaloids have already been identified in seaweed species, and algae of the genus *Caulerpa* are known sources of alkaloids with already described biological activities [98,99,100,102,116]. However, while numerous studies elucidate the anti-inflammatory activity of other seaweed compounds such as phenols and polyphenols, carotenoids, proteins and peptides, and sulfated polysaccharides, very few studies have addressed the anti-inflammatory activity of seaweed alkaloids. Further investment in the bioprospecting of these compounds is needed, and there is evidence of the unexplored anti-inflammatory potential of these molecules.

Seaweed alkaloids with distinct biological activities may have an indirect impact on inflammatory mechanisms. This is the case for alkaloids with antioxidant activity, and the antioxidant mechanisms impacting proinflammatory signaling pathways are well described in the literature [117,118]. For example, dictyospiromide (**43**), an antioxidant alkaloid obtained from the brown algae *Dictyota coriacea*, had a potent antioxidant effect on neuron-like PC12 cells, activating Nrf2/ARE signaling pathway and increasing HO-1 cell expression [119]. The Nrf2/ARE pathway regulates the expression of antioxidant and anti-inflammatory genes, inhibiting the migration of proinflammatory cells and activating a cytoprotective redox state, which is well characterized in chronic and neurodegenerative disease models [120,121]. HO-1 also participates in macrophage polarization to the M2 phenotype, inducing the expression of anti-inflammatory genes and modulating the production of proinflammatory factors such as ROS and proinflammatory cytokines [122]. Thus, alkaloids such as dictyospiromide with antioxidant activity need to be studied further in order to elucidate their anti-inflammatory potential.

PTP1B inhibitor alkaloids also have well-described biological activity, as reported in studies bioprospecting marine natural products with anti-tumor activity [123]. However, some studies have also demonstrated the participation of PTP1B as a regulator of proinflammatory signaling pathways. PTP1B -/- mice showed reduced eosinophilia in lung tissue and bronchoalveolar lavage in a mouse model of ovalbumin-induced respiratory allergy [124]. The interaction between VCAM-1 and PTP1B is necessary for ERK1/2 activation and is one of the regulatory pathways for leukocyte migration [125], which explains the observed anti-migratory effect. Inhibition of PTP1B also induces macrophage polarization to the M2 phenotype, with reduced levels of IL-1β, IL-12p70, IL-17, IL-21, IL-23, and M-CSF, in addition to increasing IL-10 production in LPS-stimulated RAW264.7 cells [126]. Alkaloids with PTP1B inhibitory activity, such as racemosin C (**40**) and caulersin (**41**), found in algae of the genus *Caulerpa*, need to be investigated as potential compounds affecting PTP1B-mediated anti-inflammatory activity.

## 5. Conclusions

Marine alkaloids with anti-inflammatory activity are compounds with great potential for pharmacological and medical use but are still a subject of bioprospecting in marine natural products that needs to be explored further. These compounds can be found in sponges, microorganisms, ascidians, and cnidaria. Very few studies have identified and characterized these molecules. Studies on seaweed alkaloids, in particular, need to be stimulated in order to elucidate the full range of biological activities of these compounds, especially their anti-inflammatory potential.

## Figures and Tables

**Figure 1 marinedrugs-18-00147-f001:**
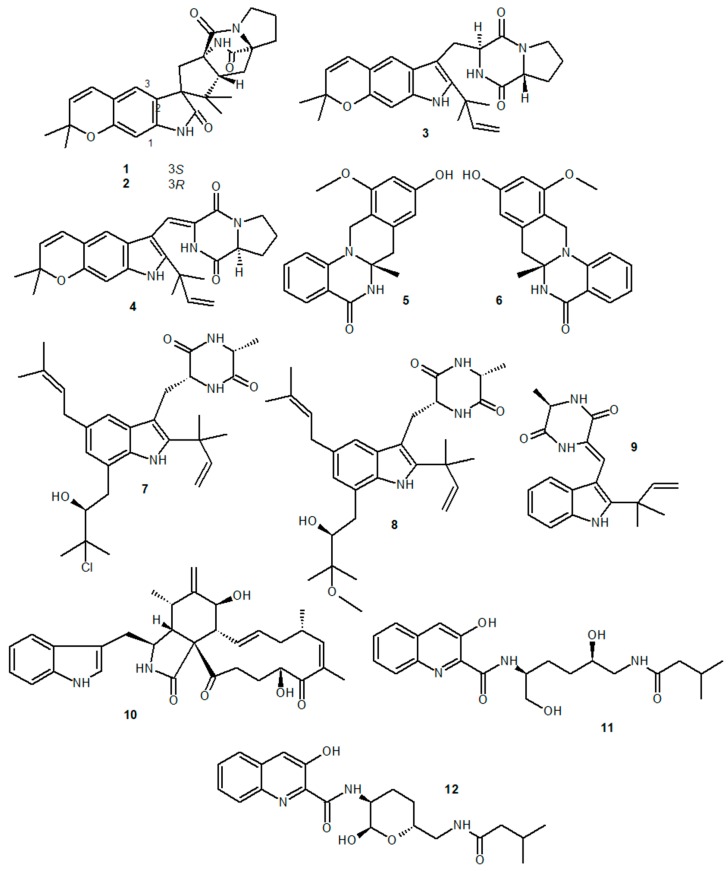
Anti-inflammatory marine alkaloids derived from fungi and bacteria.

**Figure 2 marinedrugs-18-00147-f002:**
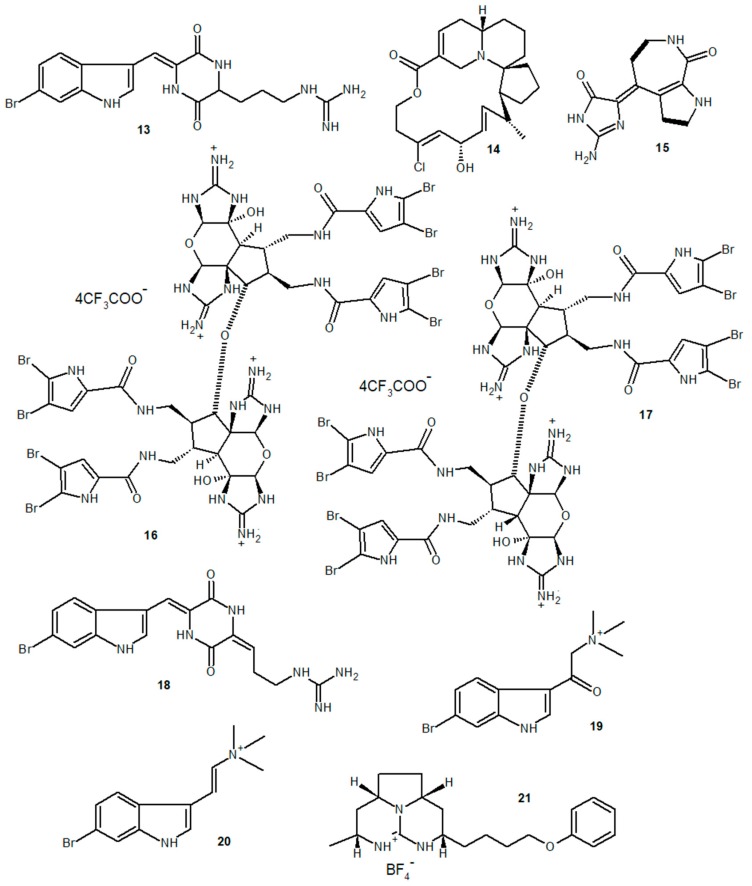
Anti-inflammatory marine alkaloids derived from sponges.

**Figure 3 marinedrugs-18-00147-f003:**
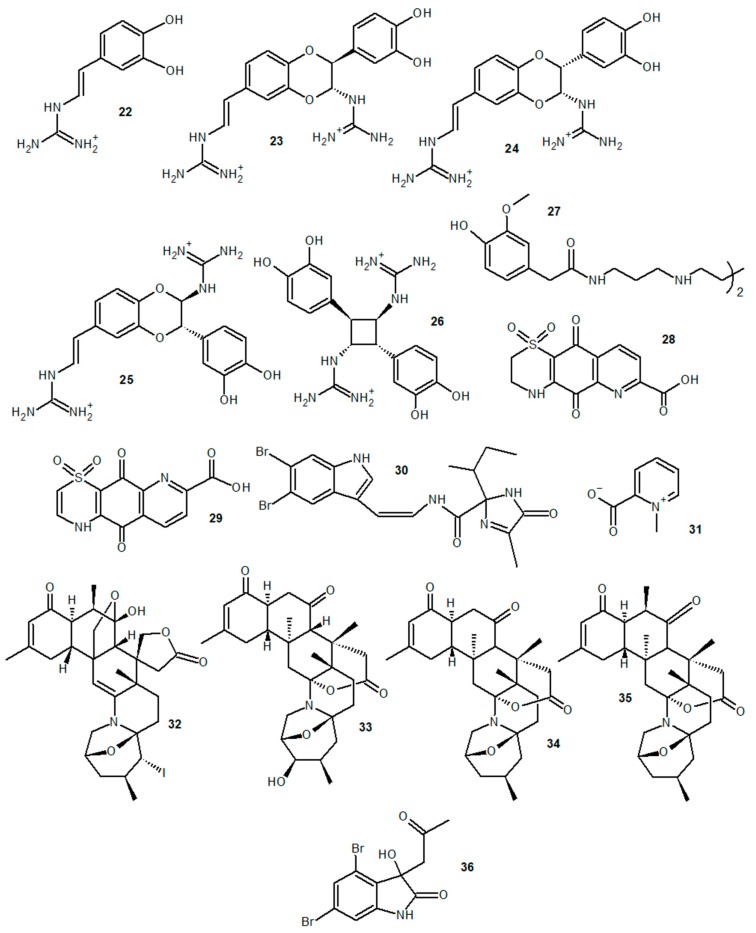
Anti-inflammatory marine alkaloids derived from ascidians, anemones, cnidarians, and bryozoans.

**Figure 4 marinedrugs-18-00147-f004:**
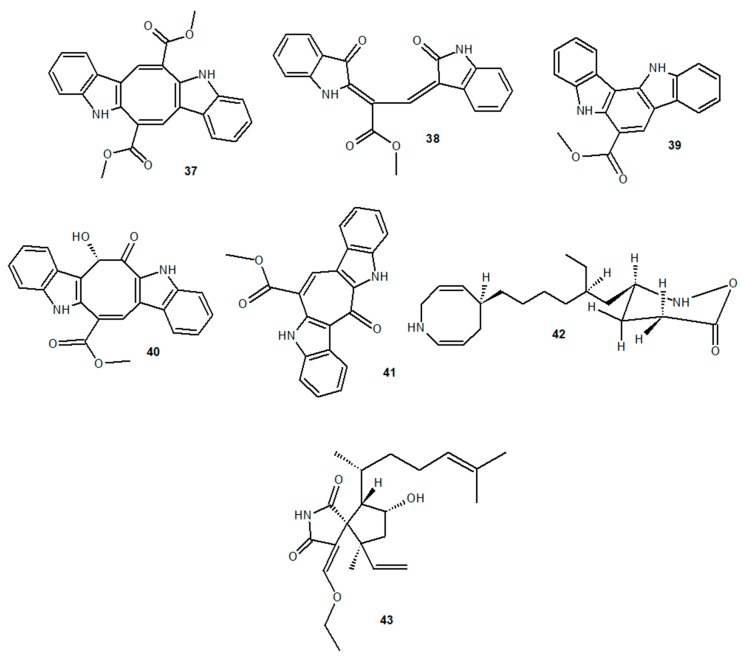
Anti-inflammatory marine alkaloids derived from algae.

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
