# Peer review of "Marine Alkaloids with Anti-Inflammatory Activity: Current Knowledge and Future Perspectives"

_marinedrugs, 2020, doi:10.3390/md18030147_

Round 1
Reviewer 1 Report
The manuscript by Souza et al. reports on a timely subject, antiinflammatory activity of marine alkaloids. Unfortunately the manuscript is poorly documented (improper, incorrect and/or missing citations) and poorly proofread (errors of style, such as use of italics and capitalization). The chemical structures are poorly rendered, making some heteroatoms hard to read, and using multiple styles (e.g., bond widths, font size, overall structure size).
Many of the attached edits can easily be corrected. However, the poor documentation will require extensive attention. For example, just because a paper in the literature says '...20% of known natural products have their pharmacological activities elucidated...' (line 32) doesn't make it true. If that paper didn't do an analysis of the rate of biological screening, it's just conjecture. It is incumbent upon the authors to find an authoritative paper that actually counted bioactive natural products. Propagating unsubstantiated data is a disservice to the community.

Author Response
Dear Reviewer 1, please follow the answers about your questions.
Coments: The manuscript by Souza et al. reports on a timely subject, antiinflammatory activity of marine alkaloids. Unfortunately the manuscript is poorly documented (improper, incorrect and/or missing citations) and poorly proofread (errors of style, such as use of italics and capitalization). The chemical structures are poorly rendered, making some heteroatoms hard to read, and using multiple styles (e.g., bond widths, font size, overall structure size).
Many of the attached edits can easily be corrected. However, the poor documentation will require extensive attention. For example, just because a paper in the literature says '...20% of known natural products have their pharmacological activities elucidated...' (line 32) doesn't make it true. If that paper didn't do an analysis of the rate of biological screening, it's just conjecture. It is incumbent upon the authors to find an authoritative paper that actually counted bioactive natural products. Propagating unsubstantiated data is a disservice to the community.
Answer: The reviewer is correct and we made the appropriate changes in the text and figures according to the notes in the manuscript reviewed, as follows:
Line 11: We deleted “already identified in the literature” as asked.
Line 19: We changed the term “bioprospection” to “bioprospecting” as suggested.
Lines 25: We changed the entire sentence to a more appropriate statement, using the correct form “ostensive” and providing additional reference. This altered the sequence of references below the text of the manuscript. These changes are now in lines 25-27.
Line 27: We changed the sentence to a more appropriate statement and provided the reference. These changes (now in lines 27-29) altered the sequence of references below the text of the manuscript.
Line 28: We changed the term “chemically” to “biologically” as suggested.
Line 30: We deleted “For decades” and changed “several” to “many,” as suggested. We also deleted “many” as asked. These changes are now in lines 29-30.
Line 33: We changed the sentence to a more appropriate statement and provided a reference for this information. This altered the sequence of references below the text of the manuscript. We also deleted “the” as asked. These changes are now in lines 31-32.
Line 36: As asked by the reviewer, we changed the reference in the sentence for a more appropriate one. This change is now in line 35.
Line 39: We provided an appropriate reference to this affirmation. This altered the sequence of references below the text of the manuscript. We also deleted the term “terrestrial” as asked by the reviewer. This change is now in line 38.
Line 40: We changed the sentence to a more appropriate statement and provided new references. This altered the sequence of references below the text of the manuscript. These changes are now in lines 38-39.
Line 40: We changed the entire sentence to a more appropriate statement and provided additional references. This altered the sequence of references below the text of the manuscript. These changes are now in lines 39-41.
Line 46: We changed to a correct reference. This change is now in line 45.
Line 49: About this question, we did not find any reference that said exactly this. When we said that “anti-inflammatory activity of compounds derived from seaweed is one of the largest bioprospecting areas in marine natural products,” we affirmed this based on our experience in the area, with more than 10 years of research in seaweed-derived anti-inflammatory substances. Also, a rapid search on PubMed using the keywords “seaweed” and all mentioned pharmacological activities cited in lines 45-48 (now 44-46) showed 94 results for “seaweed anti-inflammatory,” performing the second best result, behind only “seaweed antioxidant,” with 310 results. This led us to consider the anti-inflammatory activity of compounds derived from seaweed as “one of the largest bioprospecting areas in marine natural products,” as mentioned. Because of this, we did not add any reference in our own citation. This statement is now in lines 47-48.
Line 56: We changed the term “acid” to “acids” and deleted the term “structures” as requested. These changes are now in line 55.
Line 57: As suggested, we changed reference 23 and deleted the term “terrestrial.” These changes are now in lines 56 and 70, respectively.
Line 60: We changed the sentence to a more appropriate statement and provided additional references. This altered the sequence of references below the text of the manuscript. These changes are now in lines 70-71.
Line 60. We deleted the term “terrestrial” as asked. We also changed the sentence to add some information about known marine alkaloids, with the appropriate references, as suggested by Reviewer 2. This altered the sequence of references below the text of the manuscript. We made these changes in the sentence to adjust the text to a better form. These changes are now in lines 73-79.
Line 61: We deleted the term “various” as asked. This change is now in line 73.
Line 71: We changed the term “various” to “marine” as suggested. This change is now in line 87.
Line 73: We deleted the term “invertebrate” as asked. This change is now in line 89.
Line 74: We changed the term “several” to “many” as suggested. This change is now in line 90.
Line 76: We provided additional references for this statement. This altered the sequence of references below the text of the manuscript. Also, we deleted the term “various” as asked. These changes are now in lines 91 and 92, respectively.
Line 78: We fixed the order of the citations as asked. This change is now in line 94.
Line 85: The previous paragraph was an introductory text to the next three subsections. We mentioned fungi previously only to introduce the presence of alkaloids in fungi species already reported. In this subsection, we describe better the anti-inflammatory alkaloids reported in fungi species. We deleted the term “various” as asked. This change is now in line 99.
Line 88: We fixed the order of the citations as asked. This change is now in line 102.
Line 89: We fixed the order of the citations as asked. This change is now in line 103.
Line 89: We changed the sentence to a more appropriate statement. This change is now in line 103.
Line 90: We fixed the order of the citations as asked. This change is now in line 104.
Line 96: We changed the term “dioxopiperazine” to “diketopiperazine” as suggested. This change is now in line 110.
Line 118: We revisited and redrew the structures in Figure 1 exactly as in the cited sources, using another software to give more quality to the figure. These changes are in line 130.
Line 122: Manoalide is a sesterterpene found in marine sponges [Mar Drugs. 2010 Feb, 8(2), 313-346]. As this work focuses only on marine alkaloids, we did not include this class of molecules in the review.
Line 123: We changed the sentence to a more appropriate statement and also included data about the 2011-2017 time frame. We provided additional references for this statement. This altered the sequence of references below the text of the manuscript. These changes are now in lines 133-136.
Line 124: We deleted the term “various” as asked. This change is now in line 136.
Line 127: We provided additional references to this statement. This altered the sequence of references below the text of the manuscript. This change is now in line 139.
Line 128: This statement is an introductory sentence to the next two paragraphs. We changed the sentence to a more appropriate form. This change is now in line 140.
Line 129: We deleted the term “sponge” as asked, and changed “the” by “the sponge” as suggested. These changes are now in line 141.
Line 131: We changed “Barettin” to “barettin” as suggested. This change is now in line 143.
Line 133: We deleted the term “sponge” as requested, and changed “the” by “the sponge” as suggested. These changes are now in line 145
Line 136: We changed the term “(10Z)-debromohymenialdisine” to “(10Z)-debromohymenialdisine” as asked. This change is now in line 148.
Line 136: We changed “the” to “a” and deleted the term “species” as suggested. These changes are now in line 148.
Line 141: We deleted the term “various” as asked. This change is now in line 153.
Line 142: We changed the term “(10Z)-debromohymenialdisine” to “(10Z)-debromohymenialdisine” as asked. This change is now in line 154.
Line 146: We deleted the term “sponge” and changed “the” by “the sponge” as suggested. These changes are now in line 158.
Line 148: We deleted the term “various” as asked. This change is now in line 160.
Line 150: We deleted the term “various” as asked. This change is now in line 162.
Line 154: We changed the term “Geobarettin” to “geobarettin”. This change is in line 166.
Line 155: We changed the term “Barettin” to “barettin”. We also deleted the term “sponge” and changed “the” by “ the sponge” as suggested. These changes are now in lines 166-167.
Line 158: We changed the term “Geobarettin” to “geobarettin”. This change is now in line 169.
Line 167. We revisited and redrew the structures in Figure 2 exactly as in the cited sources, using another software to give more quality to the figure. These changes are now in line 176.
Line 171: Pseutopterosins are a group of marine diterpene glycosides found in gorgonian corals [Bioorg Med Chem. 2011 Nov, 19(22), 6702-6719]. As this work focuses only on marine alkaloids, we did not include this class of molecules in the review.
Line 174: We changed the terms “Tubastrine” and “Orthidines” to “tubastrine” and “orthidines”, respectively. These changes are now in line 182.
Line 177: We changed the terms “Turbastrine” and “Orthidine” to “tubastrine” and “orthidine”, respectively. These changes are in now line 185.
Line 179: We changed the term “sp.” to “sp.”. This change is now in line 187.
Line 182: We deleted the term “Anemone” as asked. This change is now in line 190.
Line 183: We changed the term “from” to “from the anemones”. This change is now in line 190.
Line 184: We changed the term “Homarine” to “homarine”. This change is now in line 192.
Line 189: We changed the terms “3-hydroxynorzoanthamine” and “Norzoanthamine” to “3-Hydroxynorzoanthamine” and “norzoanthamine”, respectively. These changes are now in lines 196-197.
Line 190: We changed the term “Zoanthamine” to “zoanthamine”. This change is in line 197.
Line 197: We revisited and redrew the structures in Figure 3 exactly as in the cited sources, using another software to give more quality to the figure. These changes are now in line 204.
Line 202: We deleted the term “terrestrial” as asked. This change is now in line 210.
Line 205: We changed the sentence to a more appropriate statement and provided some references. This altered the sequence of references below the text of the manuscript. These changes are now in lines 212-213.
Line 217: We changed the terms “Caulerpa” and “Caulerpin” to “Caulerpa” and “caulerpin”, respectively. These changes are now in line 225.
Line 219: We changed the term “Caulerpa” to “Caulerpa” as requested. This change is now in line 226.
Line 221: We changed the term “Caulerpin” to “caulerpin” as asked. These changes are now in line 229.
Line 222: We fixed the order of the citations as requested. We also changed the term “Caulerpa” to “Caulerpa” as asked. These changes are now in lines 229 and 230, respectively.
Line 223: We changed the terms “Racemosin” and “Caulersin” by “racemosin” and “caulersin”, respectively. These changes are now in lines 230-231.
Line 224: We fixed the order of the citations and altered the position of the citations about the alkaloids racemosins A (before line 224), B (before line 225) and C (before line 226), and caulersin (before line 226) in the text, as asked by Reviewer 2. These changes are now in line 231.
Line 227: We changed the term “Caulerpin” to “caulerpin” as asked. This change is now in line 235.
Line 230: We changed the term “Caulerpin” to “caulerpin” as asked. This change is now in line 238.
Line 231: We changed the term “Caulerpin” to “caulerpin” as asked. This change is now in line 239.
Line 234: This statement is based on Reference 67 (now Reference 102), discussed in the previous sentence. We changed the position of the reference number in the text (before in line 231) to clarify the statement. This change is now in line 243.
Line 243: We deleted all information about chromenes derived from Gracilaria sp. seaweeds contained in lines 242-251, as asked by Reviewer 2. The terms “sp.” and “algaes”, which were asked to be changed by “sp.” and deleted, respectively, were in this excluded information.
Line 264: We revisited and redrew the structures in Figure 4 exactly as in the cited sources, using another software to give more quality to the figure. These changes are now in line 261.
Line 268: We changed the term “Bioprospection” to “Bioprospecting” as suggested. This change is now in line 264.
Line 270: We changed the sentence to a more appropriate statement and provided a reference. This altered the sequence of references below the text of the manuscript. These changes are now in lines 266-267.
Line 274: We changed the sentence to a more appropriate form and provided additional references for this statement. This altered the sequence of references below the text of the manuscript. These changes are now in lines 268 and 270.
Line 277: We changed the term “Several” to “Many” as suggested by the reviewer. This change is in line 273.
Line 278: We changed the term “Caulerpa” to “Caulerpa” as asked. We also deleted the term “various” as requested. These changes are now in line 274.
Line 278: We provided additional references for this statement. This altered the sequence of references below the text of the manuscript. These changes are now in line 274.
Line 287: We fixed the order of the citations as asked by the reviewer. This change is now in line 283.
Line 287: We deleted the term “the” and changed the term “Dyctiospiromide” to “dictyospiromide”. These changes are now in line 283.
Line 288: We changed the term “brown” to “the brown” as asked. This change is now in line 283-284.
Line 292: We fixed the order of the citations. This change is now in line 288.
Line 295: We changed the term “Dyctiospiromide” to “dictyospiromide”. This change is now in line 291.
Line 306: We changed the terms “Racemosin” and “Caulersin” to “racemosin” and “caulersin”, respectively. These changes are now in line 302.
Line 340: We checked all references and fixed all citation problems of authors’ names, form, and punctuation.

Reviewer 2 Report
The authors present a review on a rather specialized topic, since they make 3 restrictions: only marine natural products, only alkaloids, only anti-inflammatory activity. Nevertheless, this is a nice and extensive peak of work, which after some revision is suitable for publication in Marine Drugs.
Concerning one of the above mentioned restrictions: The authors are encouraged to point out more clearly the evident advantages of alkaloids over non-basic drug candidates (equilibrium between neutral and protonated form, hence superior pharmacokinetic properties; option for preparation of stable, crystalline salts, …).
Introduction: I feel it was appropriate to mention the most prominent launched alkaloid drug from marine sources, trabectedin in the introduction. Another very prominent and structurally unique class of marine alkaloids, the pyridoacridines (albeit not known for anti-inflammatory activities), would deserve being mentioned as well.
Names of alkaloids need to be written in a uniform manner – I strongly recommend not to use capital letters (e.g., barettin instead of Barettin, and numerous others).
Do not present lengthy IUPAC names, nobody can comprehend these (e.g., line 251 for compound no. 45).
The list of references contains lots of mistakes: line 78: references 63, 64 most likely should be 30 and 31; in a similar manner it holds for line 89 (ref. 62 an 68 are wrong, where is ref. 35?), line 222 (94, 95 wrong, where is 61,62?), line 287 (114, 115; where is 81, 82?), line 292 (117, 118 – 84,85?). Line 223: please add references next to the substance numbers of the 4 alkaloids presented here.
Figure 3: Orthidine F (27): add a “2” at the right end of the molecule next to the parenthesis to indicate its symmetrical structure. The structures of alkaloids 33, 34, 35 are not correct! Delete unnecessary methylene protons and check the entire structures.
Line 244ff: The chromenes 42, 43, 44 are not alkaloids, they are not even well known as “precursors of alkaloids”, so these 3 compounds should be deleted from the manuscript.
Line 259, 260: please point out whether the structures of the mentioned alkaloids are known or not.
Figures: The quality of the drawn structures is rather poor in many cases, it even appears to me as if the authors included copies or screenshots of structures from other sources. Please check the figures carefully, and present all of them in a uniform style and quality.
Misprints and others: line 40: omega; line 54: delete “in the literature”; line 128: “alkaloids and alkaloid derivatives from sponges” does not make sense, since the derivatives are produced in labs, not in nature; line 129: G. barretti: present the full systematic name; line 182 and 197: Sea anemone; line 192: analogous; line 200: delete “activity”; line 287: delete “the”
Author Response
Dear Reviewer 2, please follow the answers about your questions.
Comments REVIEWER 2:
The authors present a review on a rather specialized topic, since they make 3 restrictions: only marine natural products, only alkaloids, only anti-inflammatory activity. Nevertheless, this is a nice and extensive peak of work, which after some revision is suitable for publication in Marine Drugs.
Concerning one of the above mentioned restrictions: The authors are encouraged to point out more clearly the evident advantages of alkaloids over non-basic drug candidates (equilibrium between neutral and protonated form, hence superior pharmacokinetic properties; option for preparation of stable, crystalline salts, …).
Answer: We included some information about chemical characteristics of alkaloids which made these molecules good drug candidates, as asked. We also provided appropriate references. This altered the sequence of references below the text of the manuscript. We also made changes in the paragraph to adjust the text to a better form after the inclusion of this additional information. These changes are now in lines 56-69.
Introduction: I feel it was appropriate to mention the most prominent launched alkaloid drug from marine sources, trabectedin in the introduction. Another very prominent and structurally unique class of marine alkaloids, the pyridoacridines (albeit not known for anti-inflammatory activities), would deserve being mentioned as well.
Answer: We included some information about trabectedin and pyridoacridines alkaloids as suggested. We also provided appropriate references. This altered the sequence of references below the text of the manuscript. We also made changes in the paragraph to adjust the text to a better form after the inclusion of this additional information. These changes are now in lines 73-77.
Names of alkaloids need to be written in a uniform manner – I strongly recommend not to use capital letters (e.g., barettin instead of Barettin, and numerous others).
Answer: We made the appropriate changes in all names of alkaloids in the text of the manuscript. The changes are now in lines 143, 166, 169, 182, 185, 192, 197, 225, 229, 230, 231, 235, 238, 239, 283, 291, and 302.
Do not present lengthy IUPAC names, nobody can comprehend these (e.g., line 251 for compound no. 45).
Answer: We deleted the IUPAC name for alkaloid no. 42 (before no. 45) and referred to it as “an azocinyl morpholinone alkaloid”. The number of this alkaloid changed to 42 because the deletion of compounds no. 42, 43, and 44. These changes are now in lines 250-251.
The list of references contains lots of mistakes: line 78: references 63, 64 most likely should be 30 and 31; in a similar manner it holds for line 89 (ref. 62 an 68 are wrong, where is ref. 35?), line 222 (94, 95 wrong, where is 61,62?), line 287 (114, 115; where is 81, 82?), line 292 (117, 118 – 84,85?). Line 223: please add references next to the substance numbers of the 4 alkaloids presented here.
Answer: The reviewer is correct. An error in the software used to manage the references cited in the text led to most of these mistakes and, unfortunately, we did not notice this before submitting the manuscript. We fixed the order of citations. The numbers of most citations have changed because of the reordering of wrong and missing citations and the inclusion of 36 more new references in the text, as asked by the two peer reviewers, in many points of the text. We sincerely apologize for this.
Line 78: Citations 63 and 64 were out of order. We fixed the order of citations. Now the referred statement is based on citations 51 and 52. These changes are now in line 94.
Line 89: Citations 62 and 68 were out of order, and citation 35 was missing. We fixed the order of citations. Now the referred statement is based on citations 50 and 56. These changes are now in lines 103-104.
Line 222: Citations 94 and 95 did not exist, and citations 61 and 62 were missing. We fixed the order of citations. Now the referred statement is based on citations 96 and 97. These changes are now in line 229.
Line 287: Citations 114 and 115 did not exist in the list of references, and citations 81 and 82 were missing. We fixed the order of citations. Now the referred statement is based on citations 117 and 118. These changes are now in line 283.
Line 292: Citations 117 and 118 did not exist in the list of references, and citations 84 and 85 were missing. We fixed the order of citations. Now the referred statement is based on citations 120 and 121. These changes are now in line 288.
Line 223: We altered the position of the citations about the alkaloids racemosins A (before in line 224), B (before in line 225) and C (before in line 226) and caulersin (before in line 226) in the text. These changes are now in line 231.
Figure 3: Orthidine F (27): add a “2” at the right end of the molecule next to the parenthesis to indicate its symmetrical structure. The structures of alkaloids 33, 34, 35 are not correct! Delete unnecessary methylene protons and check the entire structures.
Answer: We added the number “2” at the molecule no. 27, as suggested. Structures 33, 34, and 35 were redrawn exactly as in the cited sources. These changes are in line 204.
Line 244ff: The chromenes 42, 43, 44 are not alkaloids, they are not even well known as “precursors of alkaloids”, so these 3 compounds should be deleted from the manuscript.
Answer: We deleted all information and references about chromenes derived from Gracilaria sp. seaweeds contained in lines 242-251, as asked by the reviewer. This altered the sequence of references below the text of the manuscript. This change is in line 250 forward.
Line 259, 260: please point out whether the structures of the mentioned alkaloids are known or not.
Answer: The cited reference did not mention the structures of these alkaloids.
Figures: The quality of the drawn structures is rather poor in many cases, it even appears to me as if the authors included copies or screenshots of structures from other sources. Please check the figures carefully, and present all of them in a uniform style and quality.
Answer: We revisited and redrew all structures exactly as in the cited sources, in a uniform style, using another software to give more quality to the figures. These changes are in lines 130, 176, 204, and 261.
Misprints and others: line 40: omega; line 54: delete “in the literature”; line 128: “alkaloids and alkaloid derivatives from sponges” does not make sense, since the derivatives are produced in labs, not in nature; line 129: G. barretti: present the full systematic name; line 182 and 197: Sea anemone; line 192: analogous; line 200: delete “activity”; line 287: delete “the”.
Answer: The reviewer is correct and we made the appropriate changes, as follows:
Line 40: We changed the sentence to a more appropriate statement and deleted the term “Omega”. This change is now in lines 39-41.
Line 54: We deleted “in the literature” as asked. This change is now in line 53.
Line 128: We changed the sentence to a more appropriate form. This change is now in line 140.
Line 129: We changed the term “G. barretti” to “Geodia barretti” as suggested. This change is now in line 141.
Lines 182 and 197: We opted to use the term “anemone” only, to conciliate the suggestions of the two peer reviewers of this same statement. As we discussed only marine organisms, we consider that the lack of the term “sea” is not prejudicial to the text construction. We changed the sentence to a more appropriate form. These changes are now in line 190 and 205.
Line 192: We changed the term “analog” to “analogous”. This change is now in line 200.
Line 200: We deleted the term “activity” as asked. This change is now in line 208.
Line 287: We deleted the term “the” as asked. This change is now in line 283.
We look forward to hearing from you,
Sincerely Yours,
Janeusa Trindade Souto, PhD

Round 2
Reviewer 1 Report
This review article is much improved. The authors did a great effort to incorporate comments from the prior review.
This manuscript is a resubmission of an earlier submission. The following is a list of the peer review reports and author responses from that submission.